# Empathy and redemption: Exploring the narrative transformation of online support for mental health across communities before and after Covid-19

**Yuxuan Cai** *, **Ertong Wei**, **Xintong Cai**

Social Science Division, University of Chicago, Chicago, Illinois, United States of America

* yuxuanc@uchicago.edu

## Abstract

This study examines the impact of the COVID-19 pandemic on individuals' mental health and their online interactions, particularly within Reddit's mental health communities. By analyzing data from 15 subreddits categorized into mental health and control groups from 2018 to 2022, we observed that forums dedicated to mental health exhibited higher levels of user engagement and received more supportive responses than those in other categories. However, as the pandemic evolved, a significant decrease in online support was noted, especially within these mental health groups. This decline hints at a risk of emotional burnout among users, which poses a particularly acute challenge for individuals grappling with mental health issues. Intimate relationships have also an impact on online expression of mental health. The research underscores the pandemic's effect on online support and interaction dynamics, signaling the necessity for a deeper understanding and the development of strategies to maintain support within online communities during times of crisis.

## 1 Introduction

Covid-19 has reshaped people's communication paradigm and may have influence people's mental state. Psychologists have studied the change to people's psychologic being has been shaped by covid-19. Anxiety, one of the main evaluated subjects, has been significantly increasing in society overall during this pandemic [1–4]. As a more generic description, the Covid-19 pandemic may have intensified phycological disorders like depression, PTSD (Post-Traumatic Stress Disorder), and alcohol misuse [5].

The COVID-19 pandemic has significantly impacted people's mental state and human-computer interactions, observable across multiple dimensions, particularly through the transformation of online community discussions. Social media exposure has intensified during the COVID-19 era, notably affecting mental health. Researchers from China have found a close association between social media exposure and depression during the outbreak [6]. In our study, we aim to develop a method to explore the shifts in psychological status and supportive behavior in online communities, comparing the periods before and after the pandemic's onset

**Funding:** The author(s) received no specific funding for this work.

**Competing interests:** Disclosure of Interests. All authors have no competing interests.

by comparing the support index before and after the pandemic. This approach helps us understand the broader impacts of increased social media interaction during a global health crisis.

Social support is an important topic in mental health. Social support theory suggests that social support can buffer stress, enhance psychological resilience, and improve overall well-being. And with the increasing time people spent online, online social support is gaining significance rapidly [7–10]. We propose that the online social support of different online communities have different intensity and special characterization with regard to the inherent different of the online communities.

By combining traditional analysis and natural language processing technology and by combining large-scale text data, this study is expected to discuss the dynamic changes of people's supportive behavior in different online communities before and after the pandemic and its internal relationship.

## 2 Literature review

Online communities have been proposed to relieve people's stress and have potentials in mental health issues treatment and social support groups have been proved to provide mutual aid and support for people facing chronic diseases, threatening illness, and dependency issues [11, 12]. While the gap between people with mental illness and health care professionals has become wider, the informal groups' influence in supporting people has been gradually getting attention [9]. Although some studies suggest that participation in mental health support groups can contribute to stigma and negatively impact self-esteem, numerous empirical studies indicate that being identified with such groups can also have therapeutic effects and dual aspect of group identification underscores that while it can enhance social support and resistance to stigma, it may also adversely affect self-esteem due to the links with a stigmatized group [13–15]. Other research indicates that peer support interventions for depression lead to greater improvements in depression symptoms compared to traditional care methods, making them a valuable complement to cognitive behavioral therapy [16–19]. For people with mental diseases, the social group may even serve as a place to share information and get empathy. Based on the nature of online mental health groups, we think the content of online mental health group serves as a good indicator of people's interactions and support strategies have transformed across the pandemic.

Online communities serve different proposes, and they also have special language traits reflecting the inherent unique characteristics of each group. For example, Reddit has many different subgroups called subreddits. And it's natural to conjecture that online groups that share jokes or support Republicans have different language traits compared with mental health caring groups. Some works have already examined this topic and showed that different community has its own language style. Tran and Ostendorf investigates style and topic aspects of language in online communities, during which style is characterized by a hybrid-word and part-of-speech tag n-gram language model [20]. People support each other in online communities in different ways. Researchers have studied the mental health support and its relationship to the linguistic accommodation in online communities, showing that the language style of a post influences the level of support it receives. Sharma and De Choudhury [21] has shown that there is certain language boundary between communities, and the intensity of support in different communities are varied. The support intensity can reflect the community's own style, and serves as the traits of the specific user of this group [21].

Covid-19 has reshaped people's mental state in many ways. With a multitude of research focused on the psychological influence of Covid-19 on people's mental health, few have examined how it has influenced the online mental health support group [4]. There is already

research that examines how people's perception of being supported has changed because of Covid-19. For example, a study conducted in Sweden assessed the impact of the pandemic on neighborhood social support and social interactions. The study found that individuals from neighborhoods with high social capital reported higher levels of social interactions, emotional, and instrumental support than those from lower social capital neighborhoods, while some people feel far less supported by others in Covid-19 [22].

Past Covid-19 –interactions related research has studied how living with pandemic as a lifestyle has influenced our daily interactions in real life. Research published in Nature has analyzed that personal protective equipment, like masks, influences the comprehension of facial expressions and physical and social distancing measures [23]. However, relatively few researchers study the change of virtual communities shaped by the Covid-19, and fewer focus on how the mechanism of interactions in mental health related groups have been changed by Covid. Moreover, no previous research has yet conducted comparative analysis between different types of online information sharing groups and their difference after Covid-19.

Moreover, previous researchers studying the support intensity in online communities have been using the interactions index as a sum of the likes and comments received by the post [24–26]. Wang used the duration and frequency of online interactions to measure online interactions [27]. However, this kind of method doesn't contain the comments' emotional inclination and may falsely consider a multitude of neutral or negative comments as supportive. To fill the gap of evaluating support for others, with the employment of Large Language Models, we also introduce a new method to calculate the support index by evaluating the sentiment of the comments. Previous researcher also hasn't employed cross-community analysis, but we consider it an important part to see the specific features of different communities.

We aim to explore how the paradigm of interactions and intensity of support in online mental health communities like r/Anxiety, r/Mental illness has been changed by the Covid-19. To propose more valid research and conclusion, we also include other online information sharing communities like r/RandomKindness as comparison groups or control groups. We investigate the impacts of Covid-19 on online community's communication, in both mental health groups and other information sharing groups (noted as control groups afterwards) by conducting sentiment analysis of users' comments towards respective posts.

## 3 Research question

Leveraging prior research and our hypotheses, we develop focused research questions to analyze and compare various online information-sharing communities. Prior studies have concentrated on specific support groups, largely overlooking comparative analyses of content and support levels between different communities. At the same time, the impact of major events such as the epidemic on online support behavior also needs more exploration. So we fill the research gap by answering the following research questions:

RQ1.1: How do different online information sharing communities have different linguistic characteristic?

RQ1.2: How do different online communities have different support intensity for its followers in the group?

Second, we have specially studied the transformation of relationship-related content in online communities. The pandemic has resulted in quarantine measures and significantly increased time spent with family and partners, which may lead to tension and exacerbate mental health issues. To align with various studies regarding intimate relationship, we specifically investigate how content related to intimate relationships has changed by the pandemic, and

whether this change in content is associated with a change in the level of online support received by individuals in these online support communities.

RQ2.1: How does intimate relationship related content differ before and after the pandemic for different types of communities?

Finaly, as a horizontal comparison study, we also want to see the elements that influence the support intensity or the reply rate in other information sharing groups.

We explore the impact of Covid-19 on support dynamics within online mental health communities, drawing on findings like Charoensukmongkol's [28] that crisis conditions shift communication from support to information sharing, causing emotional exhaustion in supportive roles. Given the essential role of these forums for those seeking support, the pandemic may have altered support levels, reflecting a broader mental health decline. This variability in support, influenced by the unique character of each community, underscores the need for a comparative analysis to uncover distinct patterns. Therefore, our pivotal research question focuses on these altered dynamics:

RQ3.1: How does the Covid-19 influence people's support behaviors in mental health groups?

RQ3.2: How does Covid-19 influence people's support behaviors and other groups?

## 4 Data & method

### 4.1 Data collection and random selection

Reddit, as an online community platform, fosters user interaction around topics of interest through its unique "subreddits" sections, forming a vibrant online community [17]. Unlike other social media, Reddit promotes topic-centered dialogue in an anonymous environment, revealing communication patterns that differ from other platforms [18]. Although Reddit does not systematically collect demographic data on its users, research surveys indicate that about half of Reddit users reside in the United States. These users are generally younger, and although historically the majority have been male, the gender ratio has become more balanced in recent years [29, 30]. Moreover, Reddit serves a global audience, but the primary language of communication is English, which is particularly relevant to the scope of our study. Building on methodologies used in prior studies of online communities, we have categorized the communities into two groups. One category consists of communities related to mental health, while the control group includes communities that discuss aspects of daily life.

[31] We gathered data from five mental health-related communities and ten control group communities on Reddit, amassing a total of 5,216,505 original submission posts. These posts were sourced from Pushshift Reddit dataset which archives at The-Eye.eu/redarcs, a public data library that archives Reddit data spanning from June 2005 to December 2022, and based in accordance with reddit's policies, remove user deleted and hidden content [32, 33]. Reddit's official API is free and public and can be made available to third parties [34, 35]. Reddit allows moderators, researchers, developers, and other authorized users to access public data content for research purposes via API or third-party services [36]. In accordance with Reddit's privacy policy, developers can create software or applications, like the Pushshift Reddit API used in this study, which interact with the Reddit platform through specific requests and commands. These interactions include retrieving specific information, posting content, or performing other tasks and can see data support detail in S1 File [36, 37]. Each post contained more than one response [28]. We extracted the parent id, created time, score, and text body attributes from each comment. For submissions, we extracted the post id, title, text, created time, score,

**Interaction Measurement**

😊 **Hugging Face**

Sentiment analysis/NLP

**Online Community**

🔴 **reddit**

Reddit Data

**Self-Text Measurement**

Content Analysis

| Comment number | **Mental Group** | **Control Group** | | Posts sentiment |
|---|---|---|---|---|
| Supportive Rate | *mental_illness* *mental_health* *anxiety* *social_anxiety* *suicide_watch* | *teaching* *tales_from_retail* *writing_prompts* *writing* *random_kindness* | *jokes* *parenting* *ask_science* *get_disciplined* *you_should_know* | Posts lengh Title lengh Sorce |
| Comment Sentiment | | | | |
| Relationships | | | | |

RQ1:                                RQ2:                                RQ3:

**Cosine Similarity Detection**

N-gram

Cluster Homogeneity

Relationships Lexicon

BERT Sentiment Analysis

Time Series Analysis

**Lexicon Method**

Frequency Comparison

Ordinary Least Squares

**Large Language Model**

Sentiment Analysis

Frequency Comparison

Implement

Understand changes in language features

Understanding intimate relationships and mental health in online community

Interpret the impact of changes in the epidemic on supportive behavior

**Fig 1. Research roadmap.**

number of comments, and subreddit name, and merged each submission's content with its corresponding all comments based on their parent id. We also removed records with no comments or deleted posts, and only focus on content from January 2018 to December 2022, usernames were not examined. Given the constraints posed by data complexity and token limitations. In examining mental health-focused subreddits, a strategic sampling method was employed, collecting 20,000 posts from each targeted subreddit, culminating in an aggregate of 200,000 data entry. This contrasts with the 10,000 posts collected from each control group subreddit, enabling a refined comparative analysis. By screening for duplicate responses and removing advertising words that frequently occur, we eliminated potential bot replies and advertisements. Additionally, through filtering, we retained only posts and responses in English, resulting in a total of 188,034 posts and 1,304,575 comments for further analysis using a comprehensive framework (Fig 1). The summary table is shown as below as Table 1.

**4.1.1. Variable interpretation.** This study aims to explore the interaction between the poster and the reply, so the variables are categorized into two main groups for analysis. The 'Interaction' category includes variables such as the number of comments(reply_count), supportive rate(positive_sentiment_count), and comment sentiment score, which collectively serve to elucidate the nature of community engagement within the social platform. The 'Self' category pertains to attributes of the posts themselves, encompassing factors such as post sentiment(self_median_sentiment), post length (word_count), title length (title_word_count),

**Table 1. Communities in different groups and number of samples.**

| Control group | Selected Count | Reply Count | Total Before Cleane | After Cleaned |
|---|---|---|---|---|
| teaching | 10,000/each | 1,204,805 | 100,000 | 96,386 |
| random_kindness | | | | |
| tales_from_retail | | | | |
| writing_prompts | | | | |
| writing | | | | |
| jokes | | | | |
| parenting | | | | |
| ask_science | | | | |
| get_disciplined | | | | |
| you_should_know | | | | |
| **Supportive Group** | 20,000/each | 325,236 | 100,000 | 91,648 |
| mental_illness | | | | |
| mental_health | | | | |
| anxiety | | | | |
| social_anxiety | | | | |
| suicide_watch | | | | |

score, and the presence of relational words(contains_relational_word). Secondly, two categorical variables are included, Pandemic period(after the pandemic: 1; before pandemic: 0), group (mental group:1; control group:0).

**4.1.2. Relationship count.** We crafted a lexicon of such relational terms to measure their frequency in posts, which may significantly inform our understanding of the influence these relationships have on social interaction patterns. This lexicon includes phrases that reflect familial and intimate connections: "friend", "friends", "mother", "mom", "mum", "father", "dad", "classmate", "teacher", "cousin", "sibling", "brother", "bro", "sister", "sis", "uncle", "aunt", "grandmother", "grandma", "nana", "grandfather", "grandpa", "nephew", "niece", "partner", "spouse", "wife", "husband". The presence of these words is hypothesized to alter the dynamic of community engagement, potentially intensifying the supportive weight of discussions.

## 4.2 Employing clustering for similarity computation

Cluster technology is often used in data mining and classification, Li et al present a clustering and indexing paradigm (called Clindex) for high-dimensional search spaces to using approximate similarity searches [38]. Amer and Abdalla proposed the effectiveness of using machine learning clustering to check similarity measures for text clustering, highlighting the potential for more nuanced similarity detection beyond mere distance calculations [39]. Building on this foundation, we employed the bag of words and N-grams methods with based Tfidf-vectorizer to vectorization to analyze linguistic patterns within the dataset, concentrating on the most salient 2,000 features for each group, encompassing both 'selftext' posts and comment's 'body'. After obtaining the word features, we conducted a similarity comparison assessment of language usage between two different community groups using k-means clustering. We examined the homogeneity index results to compare the similarities between language features.

## 4.3 Calculating the support index by Bert and projection

Sentiment analysis technology, widely used in analyzing text on social media. It is a key aspect of NLP (Natural Language Processing) that helps in identifying and extracting opinions within

text data. It allows businesses, governments, and individuals to gauge online sentiments, to understand the thoughts and feelings expressed in online discourse efficiently [40]. The Hugging Face Transformers has become an effective way in machine learning for textual analysis, outperforming traditional recurrent neural networks on tasks involving the comprehension of natural language. this framework has demonstrated exceptional proficiency in sentiment analysis tasks [40, 41]. It has achieved an impressive accuracy rate of approximately 99.9% in distinguishing between positive and negative commentary at specific detection tasks [42].

For the sentiment analysis, we employed a fine-tuned BERT from Hugging Face. The model returns five probabilities for five sentiments (1 star: terrible appreciation; 2 stars: bad appreciation; 3 stars: neutral appreciation; 4 stars: good appreciation; 5 stars: excellent appreciation). For example, the fine-tuned BERT from Hugging Face returns [0.002 0.002 0.015 0.194 0.787] for 'I like you. I love you'. Then, we convert the probability distribution to a range from -1 to 1 to represent the sentiment score by applying the formula below, where -1/1 represents extreme negative/positive, and 0 stands for a neutral sentiment.

$$sentiment\ score = \frac{\left(\sum_{i=1}^{5}(Prob_i \cdot Rating_i) - 3\right)}{2}$$

Applying this formula would give the above sentence a 0.881 sentiment score. This process could ensure the model only gives one sentiment score to a sentence if the model predicts the sentence has a 100% probability of excellent appreciation. Then, we applied it to all the post's text and comments to score their sentiments. Moreover, we utilized tokenizing, adding necessary special tags, truncation, and padding to ensure that input data meets the maximum length requirements of BERT. Additionally, we accelerated computing to improve the efficiency of sentiment analysis.

## 5 Results

### 5.1 Language similarity and different support intensity in online information sharing group

We observe universal linguistic patterns across these forums but note significant differences in language and interaction strategies between mental health communities and other groups. These distinctions highlight the need for a comparative approach to fully understand the unique dynamics of each community type. Our initial results show that the language traits are remarkably consistent across different community groups, as established through clustering techniques. The similarity in language across the groups is highlighted by low homogeneity scores. Both mental health and control communities display similar linguistic patterns, as reflected in the low scores detailed below. Nonetheless, there is a noticeable, though minor, variation in the style of language between comments and the texts of posts. By applying the N-grams method using NLTK to analyze the top 2,000 features, we compared the clusters presented in Figs 2 and 3. This analysis demonstrated only slight differences in language use between comments and posts across both groups. However, the linguistic distinctions are more evident in comments than in posts, as shown in Table 2.

Despite sharing similar linguistic characteristics, we observed a marked variance in the interaction intensity among different information sharing groups. Our analysis revealed that support groups, specifically mental health groups, not only engage more frequently but also respond more positively than control groups. The general response rate to posts in support groups is approximately 0.7402, compared to 0.6716 in control groups (Table 3). Additionally, we measured the positivity of replies as an indicator of community support levels. Using BERT

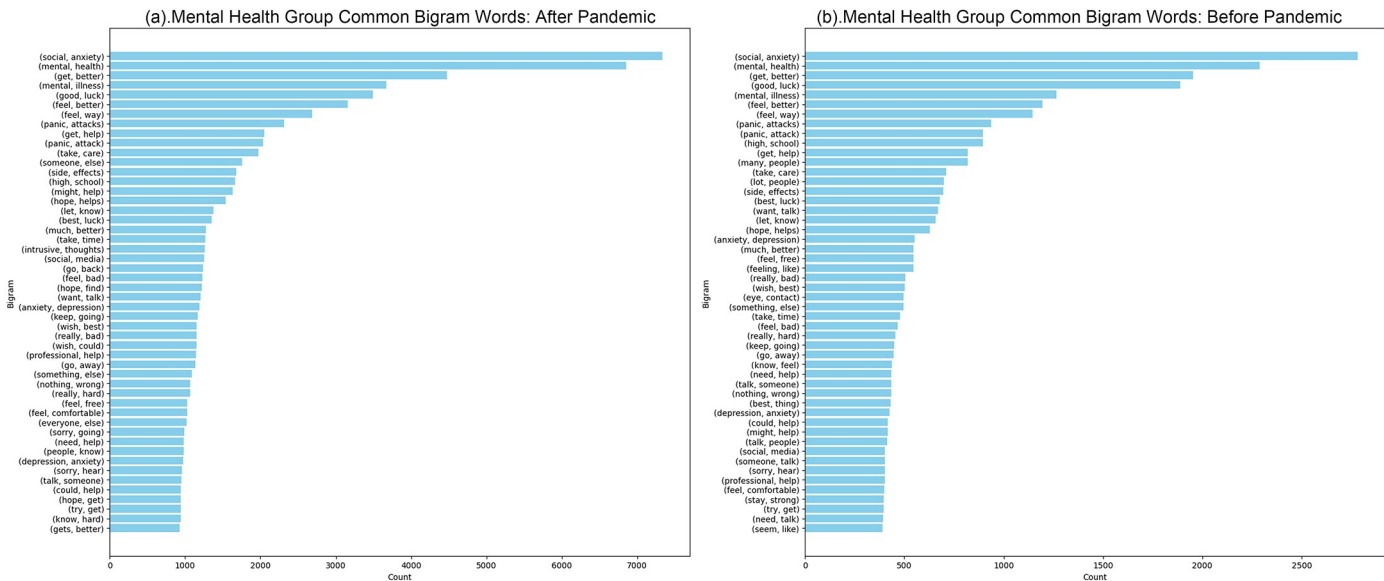

**Fig 2. Mental health group common bigram words (Top 50) before and after pandemic.** (a) Mental Health Group Common Bigram Words: After Pandemic. (b) Mental Health Group Common Bigram Words: Before Pandemic.

for sentiment analysis, we found that the positive response rate for support groups is 0.39, while for control groups, it is 0.34. These findings strongly support our hypothesis that support groups exhibit higher interaction rates than other groups.

## 5.2 Relationships and mental health

Conducting the correlation analysis, we visualize the correlation heatmap which has shown that there's a high correlation between the length of the post and whether the post is

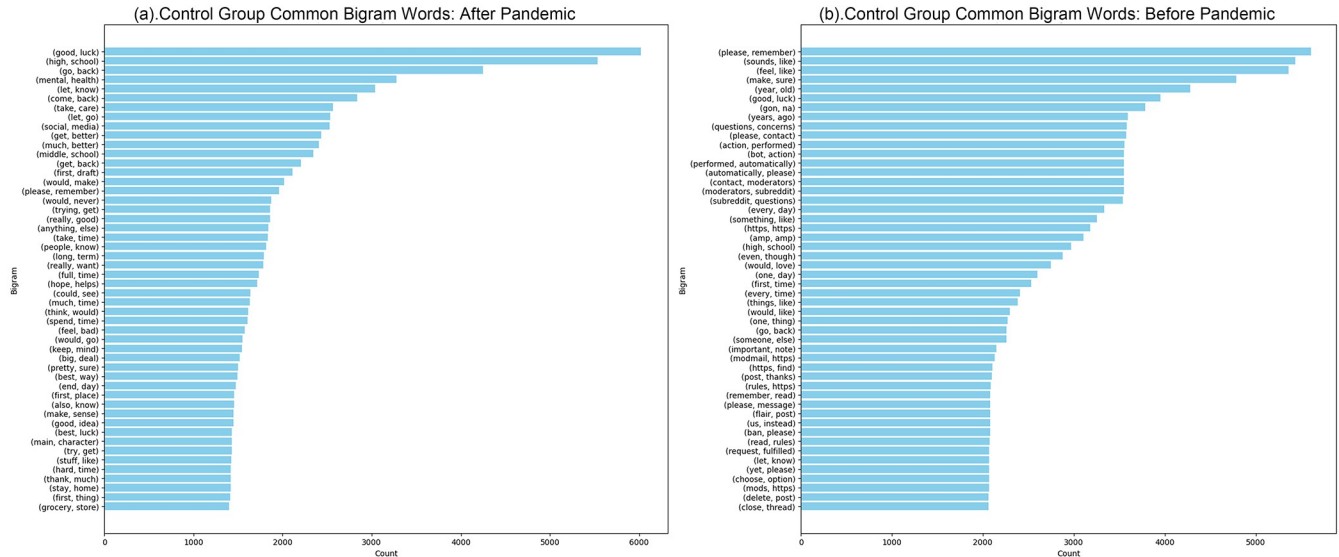

**Fig 3. Control group common bigram words (Top 50) before and after pandemic.** (a) Control Group Common Bigram Words: After Pandemic. (b) Mental Health Group Common Bigram Words: Before Pandemic.

**Table 2. Clustering results.**

| Text Group | Homogeneity | Completeness | V-measure | Adjusted Rand Score |
|---|---|---|---|---|
| Self-text | 0.042 | 0.167 | 0.067 | 0.004 |
| Comment | 0.070 | 0.195 | 0.103 | 0.014 |

**Table 3. Rate results.**

| Group name | Reply Rate |
|---|---|
| Support group | 0.7402 |
| Control Group | 0.6716 |

relationship-related content (Fig 4). The relationship related content in mental health groups is much more than the other information sharing group. Which can be shown in the table below as Table 4.

Mental health and interpersonal relationships are often deeply interconnected, almost like 'twins' in their close association. According to social support theory, this theory highlights the importance of social relationships in providing emotional and practical resources that individuals can draw upon in times of stress.

Our research has revealed that posts within mental health groups are more frequently centered on relational themes, particularly those pertaining to close relationships, such as immediate family members or intimate partnerships. This underscores a substantial proportion of mental health issues that are closely tied to relational topics.

Mental health and relationships can be closely related to each other. The relationship related content in mental health groups is much more than the other information sharing group. When the proportion of posts containing relationship words is 0.44 for supportive group and 0.24 for control groups, we can see that the relationship related content is more likely to be present in supportive groups (Table 4).

## 5.3 Diminishing empathy and convergent trend

We employ time series analysis to examine and illustrate the trajectory of the support index, revealing that mental health groups consistently exhibit a higher support index relative to control groups. This observation agrees with our initial hypothesis and corroborates findings from numerous other studies, indicating that mental health communities are more supportive and offer greater comfort to their members (Table 5).

However, as Covid-19 begins since 2020 marked as the onset of the pandemic, the support index has been increasing for both mental health groups and control groups. The mental health groups suffer from a sharper decrease of the support index, and eventually it converges to the support index trend after 2021. We calculated both the overall normalized support index and the medium normalized support index over time, and they all show the similar trend of general decreasing, with the mental health group (orange line) greatly converging to the control groups since the end of 2020 (blue line), which is especially apparent in the median picture (Figs 5 and 6).

We noticed that there is a fading digital phenomenon: Empathy in mental health groups, where people's support index has been consistently dropping, eventually converge to the control groups that is other information sharing groups. This can be interpreted that people's excessive emotional support willingness in mental health groups have been. This can also be

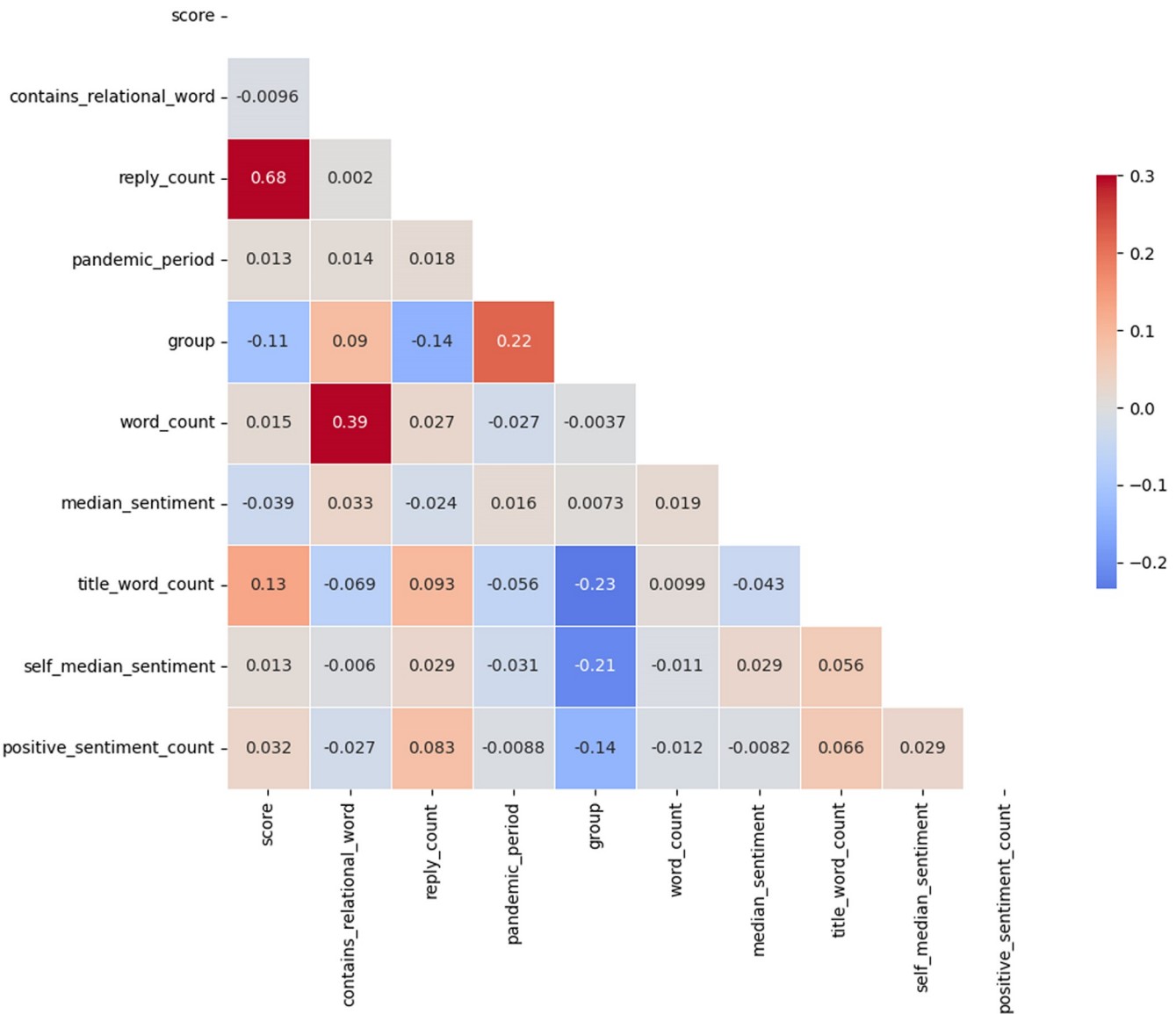

**Fig 4. Correlation matrix of all variables.**

**Table 4. Control group and mental group relationship words count and proportion.**

| Group | Include Relationship Words | Exclude Relationship Words | Proportion of Include Relationship Words |
|---|---|---|---|
| Supportive group | 43,647 | 56,353 | 0.44 |
| Control group | 24,259 | 75,741 | 0.24 |

**Table 5. Positive reply to rates of different periods.**

| Group name | Positive Reply Rate | Before Pandemic | After Pandemic |
|---|---|---|---|
| Support group | 0.388143 | 0.4293 | 0.3736 |
| Control Group | 0.341010 | 0.4174 | 0.2973 |

interpreted as that mental health patients are more sensitive to crisis like Covid-19, and therefore will give out less supportive comments to save the emotions for themes.

## 6 Discussion

Our research uses a cross-community comparative method, expanding on previous work that primarily focused on specific support communities. We examine both mental health and control groups on Reddit, aiming to explore the understudied impact of Covid-19 on online support dynamics. Specifically, we analyze how the pandemic has influenced post characteristics (such as length, theme, and sentiment) and the corresponding support levels, employing sentiment analysis over the pandemic period. We highlight the significance of intimate relationships, exacerbated by quarantine measures, in affecting mental health and online interactions. This finding aligns with previous research, indicating that psychological distress and burden

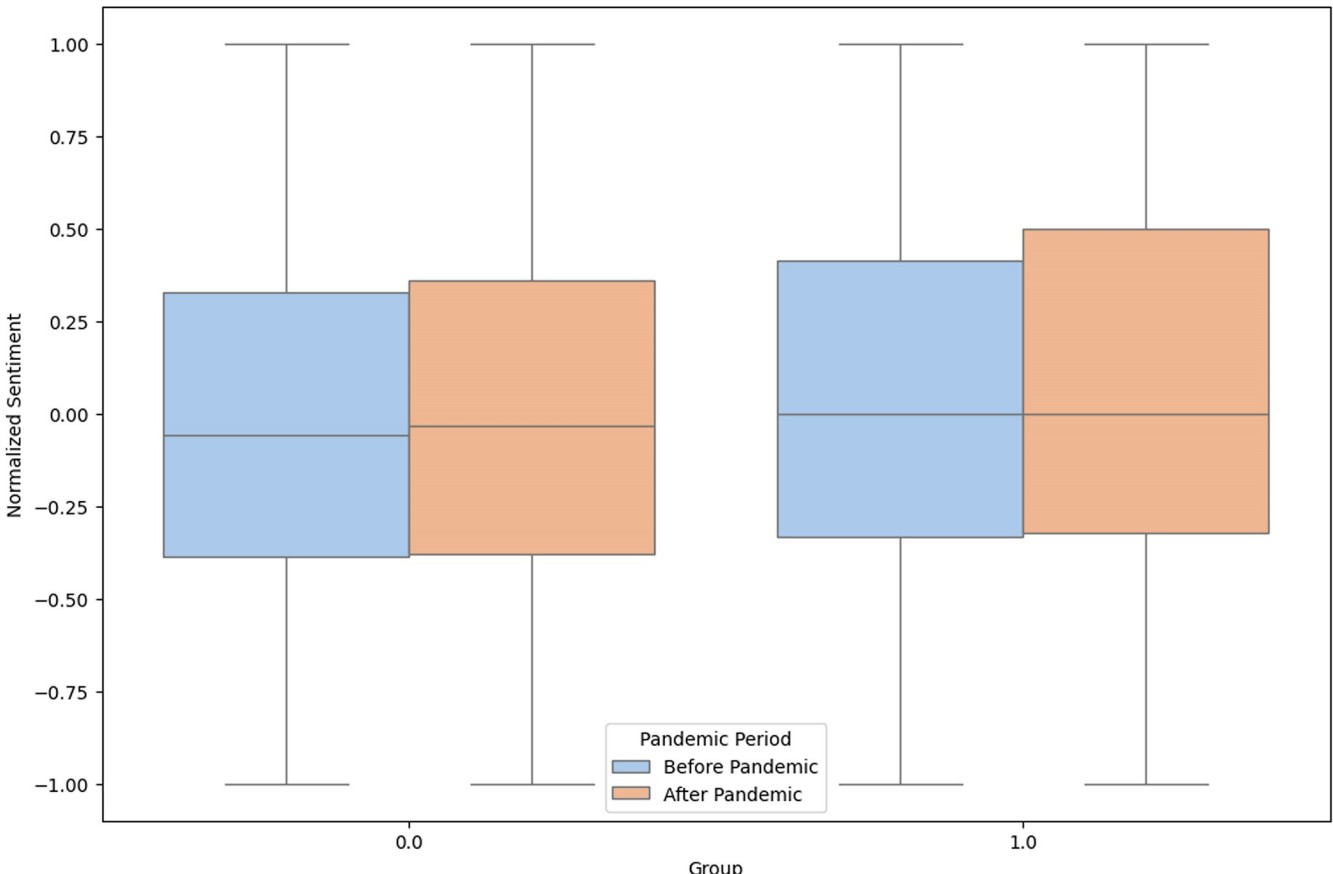

**Fig 5. Supportive rate distribution by group before and after pandemic.**

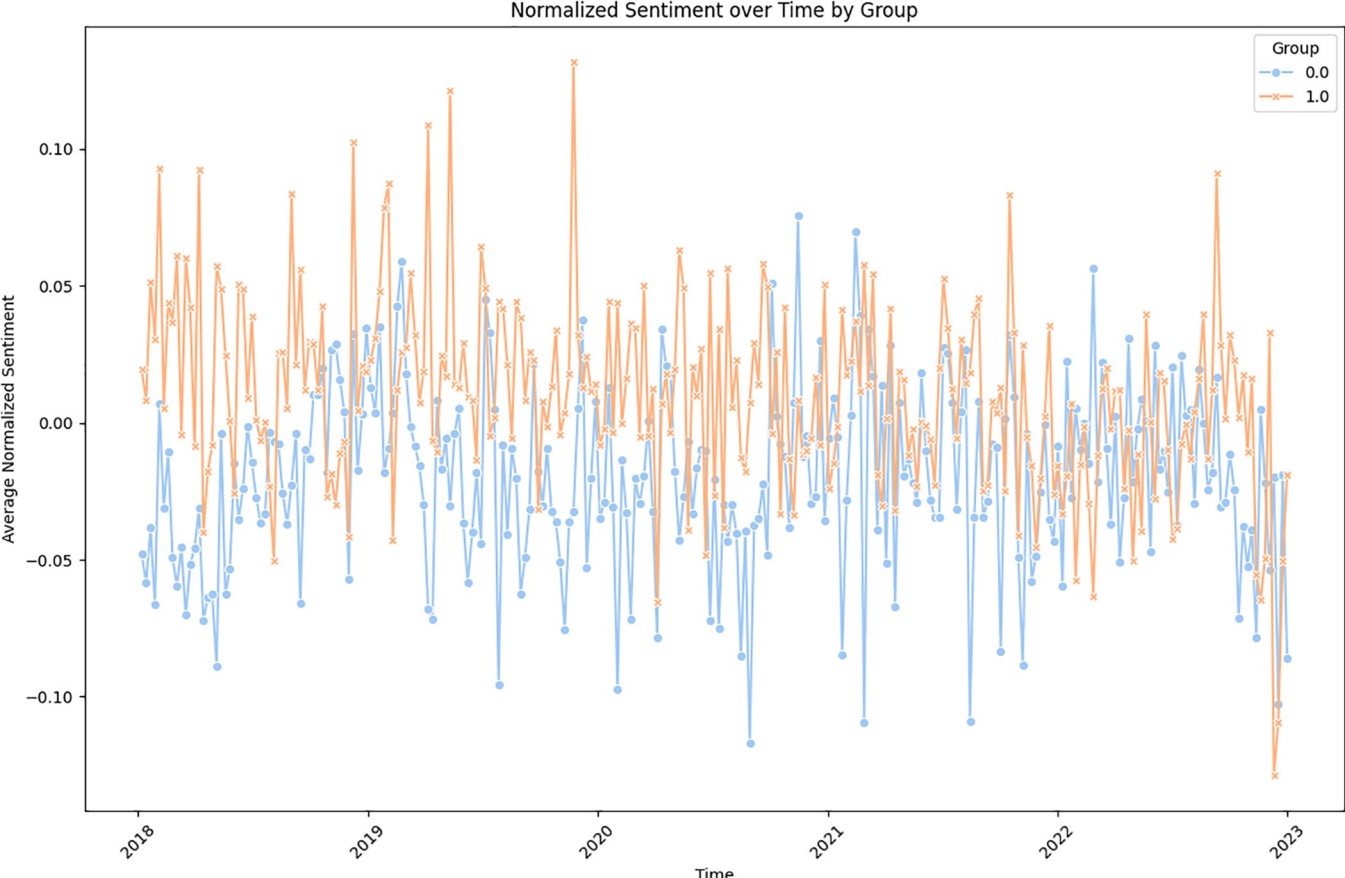

**Fig 6. Supportive rate time trend by days.**

can be alleviated through perceived social support. Specifically, support from close relationships, such as family members and friends, is significantly positively correlated with an individual's resilience [43]. Our study seeks to elucidate the evolution of discussions during the pandemic and their association with changes in online support across these communities.

In conclusion, our research indicates that as the Covid-19 pandemic progressed since 2020, the supportive index of comments within online community groups has decreased. Particularly noteworthy is the dramatic drop observed in the mental health group, eventually converging with the control groups. This suggests a diminishing willingness among individuals to offer supportive comments, reflecting a decrease in emotional support within mental health communities.

The fading digital empathy may be explained by emotional exhaustion during times of crisis or inherent characteristics of the community dynamics. According to social support theory, individuals seek and provide social support in times of need to cope with stressors and maintain wellbeing [19]. However, during crises like the Covid-19 pandemic, the availability and effectiveness of social support may diminish due to increased demands on individuals' emotional resources, disruptions in social networks, and heightened stress levels. As a result, people may be less inclined to offer supportive comments, particularly in communities focused on mental health, where the need for support may be greater.

To understand this phenomenon comprehensively, it is essential to consider a broader range of contributing factors that extend beyond our initial analysis. Platform fatigue is a significant potential factor. Previous research indicates that increased time spent on social media can impact mental well-being and reduce users' willingness to engage actively. This decrease in engagement may lead to overall reduced activity and even burnout [44–46]. Additionally, stressors within the social media environment can diminish users' sense of online control, which in turn may contribute to social media fatigue especially in the stressful environment during Covid-19 [47].

Additionally, the emergence of new forms of online support that incorporate video and audio elements might also explain shifts in user engagement. Platforms such as Discord and Zoom have risen in popularity, particularly during the Covid-19 pandemic, as they offer more direct and personal means of communication compared to text-based platforms like Reddit [48]. These platforms provide real-time audio and video interactions, which can be more appealing for users seeking more immediate and empathetic connections. The shift towards these multimedia-rich platforms might divert user attention and engagement away from traditional text-based interaction forums.

At the same time, online communication media influence how people express emotional support and the effects of such expressions. The online environment, by reducing the transmission of non-verbal cues, may change people's sympathy and support behaviors, which is particularly significant during prolonged crises. According to Walther's theory of computer-mediated communication [49], the lack of non-verbal cues common in face-to-face interactions, such as body language and facial expressions, may lead to changes in supportive behaviors. This absence could affect the way messages are received, making the emotional support expressed online susceptible to being misunderstood or perceived as insufficient.

Furthermore, research by Derks et al. [50] indicates that although electronic forms of communication like emails and instant messaging can convey emotional information, the complexity and richness of these messages are much lower than in face-to-face interactions, which may limit the depth and effect of empathetic expressions. These insights offer alternative explanations for the observed decline in engagement on Reddit.

Through in-depth analysis and cross-cultural comparisons, we hope to gain a fuller understanding of how online communities are evolving during global crises and provide evidence-based recommendations for future public health interventions and the design of online platforms to enhance emotional connection and support among users. This combination of theory and evidence has not only enriched our research discussions, but also provided valuable insight for dealing with similar crises in the future.

## 7 Limitation and future work

However, there are some limitations in our research that may be further improved in the future. Although fine-tuned BERT from Hugging Face performs well, GPT-3.5 Turbo performs better based on our observations. For instance, GPT-3.5 Turbo gives 0 (neutral) as the sentiment score of the sentence "The sentiment of this sentence is neutral", but BERT return 39% probability of terrible, and 25% probability of bad. The reason we choose BERT instead of GPT is GPT takes much longer time. We have tried to use parallel computing to make it faster, but the GPT performs worse when applied parallel computing. Another reason is the daily 500 requests limitation of GPT API. However, this is not to say the sentiment given by BERT is not accurate because BERT is good enough to meet basic emotion classification needs.

The study focuses on various communities on Reddit, and although a control group was selected, the user base of online communities is predominantly young. Based on

recommendations from media richness theory and other communication theories, future research could consider exploring this phenomenon across multiple platforms to see if there are differences between platforms [51].

Our study also proposes a future direction for research studying the effects of Covid-19 on mental state, where the willingness to support others and the capability to empathize should be an important theme. First, is there a similar result in real communities? Researchers should study more about how people with mental health illness perceive the support after pandemic and people's change in their willingness to support for others in the real neighborhoods. Communities may serve as study unit. Moreover, though the visualization and data show the fading digital empathy for group members, the exact underlying mechanism is still yet unknown. Whether it is a direct consequence of global crisis or because of social distance prolonged by quarantine remains unknown, and psychological and social science researchers should further explore the underlying mechanism for the fading digital empathy.

Another insight is that future research and policy makers should focus on the intervention to re-facilitate the support exchange process, especially for those who have relatively high demands for emotional support like mental health illness participants. For mental illness participants, compassionate interactions themselves are part of therapy and platforms should take the initiative to encourage digital empathy again.

## Supporting information

**S1 File. Data source supporting information.**
(PDF)

## Author Contributions

**Conceptualization:** Yuxuan Cai, Ertong Wei.

**Data curation:** Yuxuan Cai, Ertong Wei.

**Formal analysis:** Yuxuan Cai.

**Methodology:** Yuxuan Cai, Ertong Wei, Xintong Cai.

**Project administration:** Yuxuan Cai.

**Software:** Yuxuan Cai, Ertong Wei.

**Validation:** Xintong Cai.

**Visualization:** Ertong Wei.

**Writing – original draft:** Yuxuan Cai, Ertong Wei, Xintong Cai.

**Writing – review & editing:** Yuxuan Cai, Xintong Cai.

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
