## [Decision Letter · Decision Letter 0]

23 May 2024

PONE-D-24-14649Empathy and Redemption:  Exploring the Narrative Transformation of Online Support for Mental Health Across Communities Before and After Covid-19PLOS ONE

Dear Dr. Cai,

Thank you for submitting your manuscript to PLOS ONE. After careful consideration, we feel that it has merit but does not fully meet PLOS ONE’s publication criteria as it currently stands. Therefore, we invite you to submit a revised version of the manuscript that addresses the points raised during the review process.

We look forward to receiving your revised manuscript.

Kind regards,

Rogis Baker, Ph.D

Academic Editor

PLOS ONE

Journal Requirements:

2. In your Methods section, please include additional information about your dataset and ensure that you have included a statement specifying whether the collection and analysis method complied with the terms and conditions for the source of the data. Please also indicate whether you had access to any identifying information as part of this study. Please also check the URL provided in your Data Availability Statement is correct.

3. Please ensure that you include a title page within your main document. We do appreciate that you have a title page document uploaded as a separate file, however, as per our author guidelines (http://journals.plos.org/plosone/s/submission-guidelines#loc-title-page) we do require this to be part of the manuscript file itself and not uploaded separately.

4. Please ensure that you refer to Figure 4 in your text as, if accepted, production will need this reference to link the reader to the figure.

5. We note you have included a table to which you do not refer in the text of your manuscript. Please ensure that you refer to Table 3 in your text; if accepted, production will need this reference to link the reader to the Table.

Reviewers' comments:

Reviewer's Responses to Questions

**Comments to the Author**

1. Is the manuscript technically sound, and do the data support the conclusions?

Reviewer #1: Yes

Reviewer #2: Yes

2. Has the statistical analysis been performed appropriately and rigorously? 

Reviewer #1: Yes

Reviewer #2: Yes

3. Have the authors made all data underlying the findings in their manuscript fully available?

Reviewer #1: No

Reviewer #2: Yes

4. Is the manuscript presented in an intelligible fashion and written in standard English?

Reviewer #1: Yes

Reviewer #2: Yes

5. Review Comments to the Author

Reviewer #1: The article describes an original research method and introduces some innovations in the procedure. Online communities are an important field to research and the article offers some tools to improve that framework.

Major issues:

- There is a problem with the information structure. E.g. The authors' opinions on their own method are present in the literature review part, before the study method. Perhaps the place for this information is in the discussion part.

Another example is talking about theory and implications in the results section.

- There are some bad uses of bibliographic styles. E.g. Charoensukmongkol's (2022) or Amer and Abdalla (2020), are not written in Vancouver style.

- It is important to explain what Reddit is and how it works for non-users of this platform. It is also necessary to better describe the people who use that platform (age, countries, languages...). Some of these elements (for example, English language only) should be stated in the limitations section.

- Table 1 may contain more important information than the current one. Please explain the process of retaining and deleting posts from each group and the final result.

- The method seems not to be fully explained. There are some variables that are not explained in the methods, but are present in the results.

- Other important limitations to the decline in Reddit response engagement may be related to platform fatigue or the emergence of new forms of online support with video and audio. Consider alternative explanations for the results.

Minor issues:

- COVID-19 is written in different styles throughout the paper.

- If you use acronyms, describe their meaning the first time you use them (e.g., PSTD, NLP).

- The style of the numbers is confusing. Some of them has commas, some of them not.

- I personally have concerns about figures 2 and 3. There are many graphic decisions based on aesthetics, and not data (vertical word, word position, word color). It is certainly a very interesting graph, but not in a scientific sense.

- Take care of the details of the graphics. Lengends and captions have names like "group 0." All graphics software has ways to change these details.

- One of the most cited word in control group is “high school”, is the “teaching” community relevant to this question?

- The difference between figures 5 and 6 is not clear. They seem like two very related data with two quite similar graphs. If it is interesting to compare both results, the x-axis has to be the same (days or months).

Reviewer #2: Thank you very much for allowing me to review this research paper. I acknowledge the great effort done to finish this paper. The paper has some interesting aspects, but there are several major modifications required before it can be accepted.

- First, the introduction would benefit from providing more theoretical background on the topic. The authors should consider moving the discussion of the social support theory from the results section to the introduction, supported by scientific references.

- In the literature review, the authors should include more than one reference to support this statement "previous researchers studying the support intensity in online communities…….”..

- The in-text citations for the references to Charoensukmongkol (2022) and Amer and Abdalla (2020) are missing and need to be added.

- An abbreviation list should be included, as there are some abbreviations used in the manuscript that are not defined (e.g., NLP).

- Regarding section number 6 “Results”, I think that authors mean “Discussion” not results.

- The discussion section is currently quite superficial. The authors need to provide more detailed discussions of their results, supported by additional scientific references.

Overall, these are the major areas that require attention and modification.

Kind Regards,

6. PLOS authors have the option to publish the peer review history of their article (what does this mean?). If published, this will include your full peer review and any attached files.

Reviewer #1: No

Reviewer #2: **Yes: **Mahmoud Abdelwahab Khedr

---

## [Author Response · Author response to Decision Letter 0]

25 Jun 2024

Reviewer #1:

Reviewer #1: The article describes an original research method and introduces some innovations in the procedure. Online communities are an important field to research and the article offers some tools to improve that framework.

Major issues:

- There is a problem with the information structure. E.g. The authors' opinions on their own method are present in the literature review part, before the study method. Perhaps the place for this information is in the discussion part. Another example is talking about theory and implications in the results section.

- Thank you for pointing out these structural issues. We have revised the structure of the article accordingly. For instance, we moved the detailed discussion about our research methods to the discussion section and reorganized the literature review to align with the identified gaps and specific research questions we are addressing. Additionally, in the results section, we have restructured the discussion on theory and the details of implications, relocating them to the discussion section. We have also expanded the discussion to explore various potential reasons for the observed decline in support rates.

- There are some bad uses of bibliographic styles. E.g. Charoensukmongkol's (2022) or Amer and Abdalla (2020), are not written in Vancouver style.

- Thank you for pointing this out. We have revised the citation format to the Vancouver style as required.

- It is important to explain what Reddit is and how it works for non-users of this platform. It is also necessary to better describe the people who use that platform (age, countries, languages...). Some of these elements (for example, English language only) should be stated in the limitations section.

Thank you for your mention that point. In the limitations section, we have added explanations regarding the inherent limitations of the data and restrictions concerning user languages and we only focus on people using English. Additionally, we have detailed the Reddit platform in the methods section, including descriptions of its user demographics, usage scenarios, and mechanisms. We also validated the effectiveness of this data source based on previous studies. Further, we have discussed other research concerning this data source in the literature review section.

- Table 1 may contain more important information than the current one. Please explain the process of retaining and deleting posts from each group and the final result.

- Thank you for the suggestions. To further clarify the data processing steps, we have detailed the process of retaining and deleting posts from each group in Section 4.1, 'Data Collection and Random Selection'. The data samples were then randomly selected to ensure representativeness. In Table 1, we provide the results for each group after cleaning the posts and replies, including the number of posts and replies retained for each group and their usage in the study. We hope this information will clearly explain our data processing and selection procedures, ensuring the accuracy and reliability of our research results. We are willing to provide more detailed information if needed.

- The method seems not to be fully explained. There are some variables that are not explained in the methods but are present in the results.

- Thank you for pointing out the unclear descriptions in our methods section. We have added detailed explanations for each variable and labeled their corresponding names in the charts.

- Other important limitations to the decline in Reddit response engagement may be related to platform fatigue or the emergence of new forms of online support with video and audio. Consider alternative explanations for the results.

- Thank you for your suggestions. We acknowledge that our initial analysis may not have fully captured the complexity of factors influencing the decline in Reddit response engagement. To address this, we have expanded our discussion to include theoretical explanations and phenomena identified in related studies. We have also incorporated recent articles on changes in online platform user behavior. For example, excessive social media usage has been linked to user fatigue and burnout, which can diminish engagement behaviors. Additionally, the shifting focus towards streaming and other media may divert attention from text-based platforms, and the reduction in non-verbal cues could have impacted communication efficacy. 

Minor issues:

- COVID-19 is written in different styles throughout the paper.

- Thank you for your meticulous review and for pointing this out. We have standardized the capitalization style of "Covid-19" throughout the document.

- If you use acronyms, describe their meaning the first time you use them (e.g., PSTD, NLP).

Thank you for pointing out that we have added a detailed meaning explanation to the abbreviation.

- The style of the numbers is confusing. Some of them has commas, some of them not.

- Thank you for mentioning this, we have unified the representation of numbers.

- I personally have concerns about figures 2 and 3. There are many graphic decisions based on aesthetics, and not data (vertical word, word position, word color). It is certainly a very interesting graph, but not in a scientific sense.

- Thank you for your feedback and interest in the figures. The word cloud primarily displays the main n-gram words used by each group during and after the pandemic. By visualizing the frequency of different words through their size in the graphic. In response to your concerns regarding the scientific rigor of this approach, we have revised the presentation to include a more formal bar chart to represent these results, replacing the word cloud.

- Take care of the details of the graphics. Lengends and captions have names like "group 0." All graphics software has ways to change these details.

- Thank you for pointing out this detail. We have adjusted the legends by using graphics software.

- One of the most cited words in control group is “high school”, is the “teaching” community relevant to this question?

- Thank you for your observation. The control group indeed includes various subreddit communities such as 'teaching,' which could contribute to the frequent appearance of education-related terms like "high school." Additionally, other communities like 'parenting' and 'ask_science' may also engage in discussions related to educational topics, further influencing the presence of such terms within the group.

- The difference between figures 5 and 6 is not clear. They seem like two very related data with two quite similar graphs. If it is interesting to compare both results, the x-axis has to be the same (days or months).

- Thank you for pointing this out. Both figures indeed presented similar information, so we have removed one and replaced it with a box plot to better illustrate our findings regarding the decrease in supportive behaviors. This modification allows for a clearer visual comparison and more effective presentation of the data.

Reviewer #2: 

- First, the introduction would benefit from providing more theoretical background on the topic. The authors should consider moving the discussion of the social support theory from the results section to the introduction, supported by scientific references.

- Thank you for your suggestion. We have revised the structure of our literature review and expanded our discussion on social support theory in the introduction, supported by relevant scientific references. In addition, the introduction gives a brief summary of research objective to give readers a basic understanding of the research framework.

- In the literature review, the authors should include more than one reference to support this statement "previous researchers studying the support intensity in online communities…….”..

- Thank you for your point this out. We had added more reference for this part.

- The in-text citations for the references to Charoensukmongkol (2022) and Amer and Abdalla (2020) are missing and need to be added.

- Thank you for pointing this out. We have added the appropriate citation details in Vancouver style for these two parts.

- An abbreviation list should be included, as there are some abbreviations used in the manuscript that are not defined (e.g., NLP).

- Thank you for your suggestion. We have provided detailed definitions and explanations the first time each abbreviation appears.

- Regarding section number 6 “Results”, I think that authors mean “Discussion” not results.

- Thank you for pointing out the issue. We have corrected the section title from "Results" to "Discussion."

- The discussion section is currently quite superficial. The authors need to provide more detailed discussions of their results, supported by additional scientific references.

- Thank you for pointing this out. We have enriched the discussion section by incorporating multiple theoretical perspectives, such as computer-mediated communication, the rise of alternative media, and social media fatigue, to provide a deeper analysis of the phenomena and results observed in our study.

---

## [Editor Report · Decision Letter 1]

28 Jun 2024

Empathy and Redemption:  Exploring the Narrative Transformation of Online Support for Mental Health Across Communities Before and After Covid-19

PONE-D-24-14649R1

Dear Dr. Yuxuan Cai,

We’re pleased to inform you that your manuscript has been judged scientifically suitable for publication and will be formally accepted for publication once it meets all outstanding technical requirements.

Kind regards,

Rogis Baker, Ph.D

Academic Editor

PLOS ONE
---

## [Editor Report · Acceptance letter]

16 Jul 2024

PONE-D-24-14649R1 

PLOS ONE

Dear Dr. Cai, 

I'm pleased to inform you that your manuscript has been deemed suitable for publication in PLOS ONE. Congratulations! Your manuscript is now being handed over to our production team.

Kind regards, 

on behalf of

Dr. Rogis Baker 

Academic Editor

PLOS ONE